# PanoSent: A Panoptic Sextuple Extraction Benchmark for Multimodal Conversational Aspect-based Sentiment Analysis

Meng Luo
National University of Singapore
Singapore, Singapore
mluo@u.nus.edu

Hao Fei*
National University of Singapore
Singapore, Singapore
haofei37@nus.edu.sg

Bobo Li
Wuhan University
Wuhan, China
boboli@whu.edu.cn

Shengqiong Wu
National University of Singapore
Singapore, Singapore
swu@u.nus.edu

Qian Liu
The University of Auckland
Auckland, New Zealand
liu.qian@auckland.ac.nz

Soujanya Poria
Singapore University of Technology
and Design
Singapore, Singapore
sporia@sutd.edu.sg

Erik Cambria
Nanyang Technological University
Singapore, Singapore
cambria@ntu.edu.sg

Mong-Li Lee
National University of Singapore
Singapore, Singapore
dcsleeml@nus.edu.sg

Wynne Hsu
National University of Singapore
Singapore, Singapore
dcshsuw@nus.edu.sg

## Abstract

While existing Aspect-based Sentiment Analysis (ABSA) has received extensive effort and advancement, there are still gaps in defining a more holistic research target seamlessly integrating multimodality, conversation context, fine-granularity, and also covering the changing sentiment dynamics as well as cognitive causal rationales. This paper bridges the gaps by introducing a multimodal conversational ABSA, where two novel subtasks are proposed: 1) **Panoptic Sentiment Sextuple Extraction**, panoramically recognizing *holder, target, aspect, opinion, sentiment, rationale* from multi-turn multi-party multimodal dialogue. 2) **Sentiment Flipping Analysis**, detecting the dynamic sentiment transformation throughout the conversation with the causal reasons. To benchmark the tasks, we construct PanoSent, a dataset annotated both manually and automatically, featuring high quality, large scale (10,000 dialogues), multimodality (text, image, audio and video), multilingualism (English, Chinese and Spanish), multi-scenarios (over 100 domains), and covering both implicit&explicit sentiment elements. Further, to effectively address the tasks, we devise a novel Chain-of-Sentiment reasoning framework, together with a novel multimodal large language model (namely Sentica) and a paraphrase-based verification mechanism. Extensive evaluations demonstrate the superiority of our methods over strong baselines, validating the efficacy of all our proposed methods. The work is expected to open up a new era for the ABSA community, and thus all our codes and data are open at https://PanoSent.github.io/.

---

*Hao Fei is the corresponding author.

*MM '24, October 28-November 1, 2024, Melbourne, VIC, Australia*
© 2024 Copyright held by the owner/author(s).
ACM ISBN 979-8-4007-0686-8/24/10
https://doi.org/10.1145/3664647.3680705

## CCS Concepts

• **Computing methodologies → Atificial Intelligence**.

## Keywords

Sentiment Analysis, Multimodal Learning, Large Language Model

**ACM Reference Format:**
Meng Luo, Hao Fei, Bobo Li, Shengqiong Wu, Qian Liu, Soujanya Poria, Erik Cambria, Mong-Li Lee, and Wynne Hsu. 2024. PanoSent: A Panoptic Sextuple Extraction Benchmark for Multimodal Conversational Aspect-based Sentiment Analysis. In *Proceedings of the 32nd ACM International Conference on Multimedia (MM '24), October 28-November 1, 2024, Melbourne, VIC, Australia.* ACM, New York, NY, USA, 10 pages. https://doi.org/10.1145/3664647.3680705

## 1 Introduction

The quest for human-level artificial intelligence encompasses not only possessing intelligence but also understanding human emotions, thus propelling sentiment analysis and opinion mining to become the key area of research focus. Through decades of research, sentiment analysis has seen significant developments across various dimensions and aspects [7, 54, 58]. The field has evolved from traditional coarse-grained analysis, such as document and sentence-level analysis [71, 85], to fine-grained one (e.g., ABSA) [59, 66, 91], incorporating a wide array of emotional elements and evolving to extract different sentiment tuples, including the extraction of *targets, aspects, opinions*, and *sentiments*. Moreover, the sentiment analysis scope has broadened from purely textual content to multimodal content such as images and videos [24, 32, 40, 41, 49, 86]. Such expansion recognizes that in real-world scenarios, users often convey their opinions and emotions more accurately through diverse multimedia, providing additional information beyond text, such as micro-expressions, tone of voice, and other cues. Additionally, research has expanded beyond single-text scenarios to consider more complex conversational contexts [38, 95], where individuals frequently engage in multi-turn, multi-party discussions on social

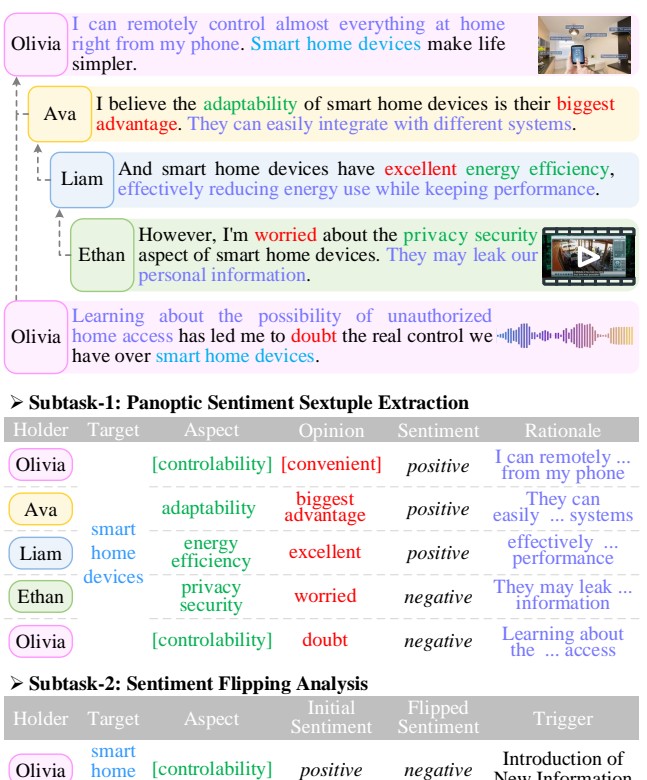

> **Subtask-1: Panoptic Sentiment Sextuple Extraction**

| Holder | Target | Aspect | Opinion | Sentiment | Rationale |
|---|---|---|---|---|---|
| Olivia | | [controlability] | [convenient] | *positive* | I can remotely ... from my phone |
| Ava | smart home devices | adaptability | biggest advantage | *positive* | They can easily ... systems |
| Liam | | energy efficiency | excellent | *positive* | effectively ... performance |
| Ethan | | privacy security | worried | *negative* | They may leak ... information |
| Olivia | | [controlability] | doubt | *negative* | Learning about the ... access |

> **Subtask-2: Sentiment Flipping Analysis**

| Holder | Target | Aspect | Initial Sentiment | Flipped Sentiment | Trigger |
|---|---|---|---|---|---|
| Olivia | smart home devices | [controlability] | *positive* | *negative* | Introduction of New Information |

**Figure 1: Illustration of the PanoSent benchmark. In [*] are the implicit elements that should be inferred from contexts.**

media platforms (e.g., Twitter, Facebook) about services, products, sports, etc.

Despite significant progress, current research definitions of sentiment analysis are still not comprehensive enough to offer a complete and detailed emotional picture, primarily due to several issues. **First**, there is a lack of an integrated definition that combines fine-grained analysis, multimodality, and conversational scenarios. In real-life applications, such as on social media and forums, these aspects often need to be considered together. However, existing studies either lack detailed analysis in multimodal sentiment analysis definitions [51, 68] or miss multimodal modeling in conversational ABSA [27, 37]. The most complete text-based ABSA definitions still do not fully cover or finely detail the granularity of emotional elements. **Second**, current sentiment analysis definitions only consider identifying fixed static emotional polarities [3, 8], neglecting the dynamic nature of emotions that change over time or due to various factors. For example, a person's original opinion in a social media conversation may change after being exposed to new information or viewpoints from other speakers. **Third**, and most critically, existing work has not thoroughly analyzed or identified the causal reasons and intentions behind sentiments [55, 57]. The arousal and change of human emotions have specific triggers, and failing to understand the causal rationale behind emotions from a cognitive perspective means that human-level emotional intelligence has not been fundamentally achieved. Overall, providing a more comprehensive sentiment analysis definition could significantly enhance the practical value of this task, e.g., developing

smarter voice assistants, better clinical diagnostic and treatment aids, and more anthropomorphic customer service systems.

To fill these gaps, this paper proposes **Multimodal Conversational Aspect-based Sentiment Analysis**, where we aim to provide a more comprehensive and holistic ABSA definition that includes both **Panoptic Sentiment Sextuple Extraction** (subtask-I) and **Sentiment Flipping Analysis** (subtask-II), as exemplified in Figure 1. Our focus is on conversational scenarios covering the four most common modalities for emotional expression in daily life, i.e., *text, image, audio, video*. On the one hand, we extend the current ABSA quadruple extraction definition to sextuple extraction, including *holder, target, aspect, opinion, sentiment*, and *rationale*, fully covering finer-grained emotional elements to offer a panoramic view of sentiment. On the other hand, we define a task to monitor the dynamic sentiment change towards the same target and aspect by the same holder throughout the conversation, and also identify the trigger reasons behind these flipped sentiments. For both sextuple extraction and sentiment change identification, we also emphasize discerning the underlying causal rationale or trigger, striving to not only know how but also why from a cognition perspective.

To benchmark the novel task, we accordingly construct a large-scale high-quality dataset, **PanoSent**. PanoSent covers more than 100 common domains and scenarios, which, based on multi-turn and multi-party conversational contexts, the sentiment elements within a sextuple may cross utterances. To mimic real human emotional expression habits, where 1) elements can originate from both textual and non-textual (audio or visual) modalities, and 2) emotions may be expressed implicitly, the data covers both implicit and explicit sentiment elements. To ensure the benchmark generalizability, the dataset includes three mainstream languages: English, Chinese, and Spanish. We collect the data from real-world sources, carefully annotated manually. To enlarge the quantity, we further automatically synthesize the dataset via OpenAI GPT-4 [1] with multimodal retrieval. Strict human inspection and cross-validation ensure high-quality standards. In total, we obtain 10,000 annotated dialogues for PanoSent.

Compared to existing ABSA tasks, the new task proposed in this work poses greater challenges, such as the need to understand complex conversational contexts and flexibly extract features from various modalities, especially discerning causal reasons at a cognitive level. Considering the recent great successes of Multimodal Large Language Models (MLLMs) in powerful semantic understanding across multiple modalities [23, 42, 46, 76], we construct a backbone MLLM system, **Sentica**, for encoding and understanding multimodal conversational content. Inspired by the human process of sentiment analysis, we further develop a Chain-of-Sentiment (CoS) reasoning framework for a high-performing task solution, which, based on the Chain-of-Thought [72] idea, breaks down the task into four progressive reasoning steps, from simpler to more complex. The system allows to more effectively extract the elements of the sentiment sextuple and identify flipped sentiments step by step, while simultaneously inducing the corresponding rationale and triggers. Furthermore, a paraphrase-based verification (namely PpV) mechanism is introduced to enhance the robustness of the CoS reasoning process. We conduct extensive evaluations on the PanoSent dataset covering two subtasks and three languages. The

**Table 1: Summary of existing popular benchmarks of sentiment analysis (representatively summarized, not fully covered).**

| Benchmark | Granularity | Sentiment Picture | Modality | Scenario | Language | Causal Rationale | Sentiment Change |
|---|---|---|---|---|---|---|---|
| CR [4] | Coarse | Sentiment | Text | Sentence | EN | ✗ | ✗ |
| Yelp [70] | Coarse | Sentiment | Text | Document | EN | ✗ | ✗ |
| SemEval [62] | Fine | Target, Aspect, Sentiment | Text | Sentence | EN | ✗ | ✗ |
| TOWE [16] | Fine | Aspect, Opinion | Text | Sentence | EN | ✗ | ✗ |
| ACOS [6] | Fine | Target, Aspect, Opinion, Sentiment | Text | Sentence | EN | ✗ | ✗ |
| ASTE [61] | Fine | Aspect, Opinion, Sentiment | Text | Sentence | EN | ✗ | ✗ |
| DiaASQ [37] | Fine | Target, Aspect, Opinion, Sentiment | Text | Dialogue | EN, ZH | ✗ | ✗ |
| Twitter2015 [50] | Fine | Target, Sentiment | Text, Image | Sentence | EN | ✗ | ✗ |
| CMU-MOSEI [87] | Coarse | Sentiment | Text, Audio, Video | Sentence | EN | ✗ | ✗ |
| IEMOCAP [5] | Coarse | Sentiment | Text, Audio, Video | Dialogue | EN | ✗ | ✗ |
| MELD [63] | Coarse | Sentiment | Text, Audio, Video | Dialogue | EN | ✗ | ✗ |
| M3ED [93] | Coarse | Sentiment | Text, Audio, Video | Dialogue | ZH | ✗ | ✗ |
| **PanoSent** | Fine | Holder, Target, Aspect, Opinion, Sentiment, Rationale | Text, Image, Audio, Video | Dialogue | EN, ZH, SP | ✓ | ✓ |

results demonstrate that our method outperforms strong LLM-based baselines, validating the efficacy of the proposed mechanisms, i.e., Sentica, CoS and PpV. Further comprehensive analyses are shown for a better understanding of all our proposals.

In summary, this work makes three significant contributions:

- For the first time, we thoroughly upgrade ABSA with a more comprehensive definition at the cognitive level, Multimodal Conversational Aspect-based Sentiment Analysis, introducing Panoptic Sentiment Sextuple Extraction and Sentiment Flipping Analysis tasks, achieving the ultimate form of sentiment analysis within the community.
- We contribute a large-scale, high-quality benchmark dataset, PanoSent, featuring multiple aspects: conversational contexts, multimodality, multilingualism, and multidomain.
- We propose an advanced reasoning framework, the Chain-of-Sentiment, based on our Sentica MLLM, achieving high task performance and providing a strong baseline for subsequent research on PanoSent.

## 2 Related Work

This work majorly focuses on the track of ABSA [10, 90]. ABSA has evolved from its initial objective of identifying sentiment polarity to more complex tasks such as recognizing targets, aspects, and opinions [33, 43, 47]. The complexity of ABSA tasks has increased with the introduction of combinations of these elements, ranging from paired extraction [9, 78] to triplet [52, 61] and quadruple extractions [6, 37]. Concurrently, multimodal SA [30], a pivotal topic within the multimodal research community [19, 20, 77, 82], has garnered increasing attention, incorporating modalities beyond text, such as images, audios, and videos. The trend in multimodal sentiment analysis has shifted from coarse-grained to fine-grained. The proposed methods mainly focus on exploring feature extraction and fusion from diverse modal inputs [23, 29, 45, 74, 86, 94], relying on additional structured knowledge [19, 21]. Furthermore, in terms of application scenarios, there has been a shift from analyzing single pieces of text to engaging in multi-turn, multi-party dialogues [88, 92], aiming to recognize emotions within dialogues to better align with real-world applications. Subsequently, dialogue sentiment analysis has gradually evolved into dialogue ABSA [37], incorporating non-textual modalities in the analysis.

However, we find that current ABSA benchmarks still lack a combined perspective and comprehensive definition across granularity, multimodality, and dialogue contexts. For instance, there is an absence of benchmarks for fine-grained sentiment analysis

in multimodal dialogue scenarios [59, 91]. Regarding granularity, there is potential to go beyond the four elements of target, aspect, opinion, and sentiment, to include the consideration of the sentiment holder, which also plays a pivotal role in a dialogue context. Moreover, previous research has not fully leveraged the role of multimodality in ABSA. In most cases, multimodal information is merely considered as supplementary clues to assist in determining opinions or sentiments [53, 67], with most of the other elements (e.g., targets, aspects) coming from texts. However, we argue that multimodality can also serve as a crucial source of information for the implicit identification of all elements more than sentiment. For example, a 'cellphone' may not be mentioned in the utterance, but the image showing a phone might feature it as the 'target'12 element. Beyond that, two other key aspects have not been sufficiently addressed in the existing ABSA. First, the dynamic nature of sentiments, especially within the context of dialogues, has not been explored. Second, the cognitive causes and intentions behind sentiments have been overlooked. In response, this work introduces a new benchmark, PanoSent, aiming to bridge all the above gaps, and provide a platform for the next phase of more comprehensive and in-depth ABSA research. Table 1 summarizes the key differences between ours and existing benchmarks.

Beyond contributing new data, we also propose an advanced methodology for this benchmark. We take full advantage of the significant success of existing MLLM [22, 75, 83, 89] in understanding multimodal data. To address the challenges posed by the new tasks, which rely on cognitive-level reasoning, we introduce a novel reasoning framework, CoS. Inspired by the existing CoT strategy, which breaks down the problem into smaller chained steps for step-by-step resolution [17, 73], we decompose the two tasks in PanoSent, significantly enhancing the task-solving efficacy. Overall, our new benchmark data and methods are poised to open up a new era for the ABSA community.

## 3 Task Definition

We formally give the definitions of two subtasks, which also are illustrated in Figure 1 with specific examples.

**Subtask-I: Panoptic Sentiment Sextuple Extraction.** Given a dialogue $D = \{u_1, \ldots, u_n\}$ with the replying structure $\{(u_i, u_j), \ldots\}$ (i.e., $u_i$ replies to $u_j$), the task is to extract all sextuples $(h, t, a, o, s, r)$. Each utterance $u_i = \{w_1, \ldots, w_{m_i}\}$ contains $m_i$ words in the text (denoted as $I^t$), occasionally with associated non-text information piece, i.e., image ($I^i$), audio ($I^a$), video ($I^v$). The elements $h$ (holder), $t$ (target), $a$ (aspect), $o$ (opinion), and $r$ (rationale) can be either the

**Table 2: Main statistics of PanoSent dataset.**

| | | Dialogue | | | Sextuple | | Modality | | | | | Manner | |
|---|---|---|---|---|---|---|---|---|---|---|---|---|---|
| | | Dia. | Utt. | Spk. | Sext. | Flip. | Txt. | Img. | Aud. | Vid. | Mix. | Imp. | Exp. |
| EN | Total | 6,000 | 28,822 | 26,831 | 28,464 | 2,136 | 3,360 | 1,320 | 360 | 240 | 720 | 1,680 | 4,320 |
| | Real | 2,000 | 9,573 | 8,827 | 9,298 | 694 | 1,102 | 427 | 108 | 70 | 232 | 536 | 1,464 |
| | Synth | 4,000 | 19,249 | 18,004 | 19,166 | 1,442 | 2,258 | 893 | 252 | 170 | 488 | 1,144 | 2,856 |
| ZH | Total | 3,000 | 14,033 | 13,444 | 13,965 | 1,068 | 1,680 | 660 | 180 | 120 | 360 | 840 | 2,160 |
| | Real | 1,000 | 4,702 | 4,510 | 4,672 | 360 | 582 | 210 | 63 | 41 | 125 | 289 | 711 |
| | Synth | 2,000 | 9,331 | 8,934 | 9,293 | 708 | 1,098 | 450 | 117 | 79 | 235 | 551 | 1,449 |
| SP | Total | 1,000 | 4,667 | 4,490 | 4,671 | 356 | 560 | 220 | 60 | 40 | 120 | 280 | 720 |
| | Real | 333 | 1,547 | 1,488 | 1,551 | 114 | 181 | 72 | 18 | 12 | 35 | 90 | 243 |
| | Synth | 667 | 3,120 | 3,002 | 3,120 | 242 | 379 | 148 | 42 | 28 | 75 | 190 | 477 |
| | All | 10,000 | 47,522 | 44,765 | 47,100 | 3,560 | 5,600 | 2,200 | 600 | 400 | 1,200 | 2,800 | 7,200 |

continuous text spans explicitly mentioned in utterances, or implicitly inferred from contexts or non-text modalities. $s$ represents the sentiment category (positive, negative, or neutral).

**Subtask-II: Sentiment Flipping Analysis.** Given input $D$, the same as in subtask-I, the task detects all sextuples $(h, t, a, \zeta, \phi, \tau)$. Here, $h$, $t$, and $a$ denote the holder, target, and aspect, consistent with the definitions in subtask-I. $\zeta$ and $\phi$ represent the initial and flipped sentiments, respectively, highlighting the dynamic change in sentiment by the same speaker towards the same aspect of the same target. $\tau$ refers to a trigger that induces the sentiment transition, which is a pre-defined label among four categories: 1) *introduction of new information*, 2) *logical argumentation*, 3) *participant feedback and interaction*, and 4) *personal experience and self-reflection*. Since subtask-II shares multiple elements with subtask-I, it is natural to detect the flipping based on the results from subtask-I to minimize redundancy.

## 4 New benchmark: PanoSent

Here we elaborate on the construction of the new dataset for multimodal conversational ABSA, as well as its key characteristics.

### 4.1 Dataset Construction

**Constructing via Human Annotation.** The corpus of dialogues is collected by crawling via publicly available APIs from various social media or forum platforms in different languages, such as Twitter, Facebook, Reddit, Weibo, Xiaohongshu, BeReal, and more. While the majority of these dialogues are text-based, some also include multimodal interactions. Then, we conduct a rigorous screening process (via both manual inspection and automated filters, e.g., keyword and Toxic-BERT detection[1]), to eliminate content (e.g., multimodal information) or instances that are harmful, private or unrelated to the dialogue. After obtaining a cleansed corpus, we commence the annotation of aspect-based sentiment sextuples. We stick to the SemEval guidelines [62] and customize the annotation manual to accommodate both subtasks of our benchmark. We recruit annotators, training them according to the manual. To guarantee reliability, each dialogue is annotated independently by at least three distinct annotators. After annotation, we calculate the Cohen's Kappa score [12], achieving a score of **0.85**, which reflects the high quality of our annotated dataset. In instances with inconsistent annotations, linguists and native speakers will collaboratively determine the final annotation. For unresolved ambiguities, the instances will be dropped.

**Constructing via Auto-Synthesis.** We find the cost and workload in the above manual annotation process to be significantly high.

---

[1]https://github.com/unitaryai/detoxify

The key issue is that real-world data sources that can provide a sufficient data volume meeting our task definition (to cover various modalities) are very rare. Hence, we consider automating data synthesis to substantially expand the volume, with the basic idea of '*automatic synthesis + multimodal retrieval*'. We first leverage the powerful LLMs for synthesizing dialogues and sextuples. A considerable amount of existing related work [15, 56, 60] has already demonstrated that OpenAI's GPT-4 can generate data of very high quality that almost perfectly matches the real distribution. Specifically, following the prior practices [15, 79], we prepare template prompts to guide GPT-4 to generate pseudo-dialogues, along with sextuple and flipping annotations. Besides, for a portion of dialogue utterances, we also instruct GPT-4 to create appropriate captions as the image, audio, and video placeholders, according to the contexts.

With the annotated dialogues, we now use the captions to retrieve the piece of information in the corresponding modality (image, audio or video) from the external multimodal databases, with only the top-10 retrieved candidates kept. Specifically, we consider multiple large-scale databases, including COCO [44], Flickr30k [84], AudioSet [25], WaveText5K [13], and WebVid [2], etc. Also we consider direct retrieval from the Google search engine, to ensure comprehensive coverage. For the associated multimodal contents, three annotators will assign a ranking score (1-10) to the 10 candidates, which are further ranked via their averaged scores, and the highest-scored one is elected as the determined multimodal information piece. Finally, each synthesized dialogue, the annotations of two subtasks, and the multimodal contents will be thoroughly examined by at least two workers. All the possibly problematic instances will be dropped. We also calculate the Cohen's Kappa score across workers, achieving a score of **0.82**, ensuring a high consistency of the synthesized annotations.

### 4.2 Data Insights

We select a portion of the real data to serve as developing and testing sets, while the remainder of the real data and all the synthesized data are used as the training set. Ultimately, the ratio of the train/dev/test sets for each language is 8:1:1. Following we briefly summarize the key characteristics and highlights of our PanoSent dataset.

**Panoptic Fine-grained Sentiment Definition.** In contrast to existing ABSA datasets, such as TOWE [16], ASTE [61], and DiaASQ [37], PanoSent dataset encompasses the most comprehensive elements, featuring six key items for ABSA.

**Cognitive Causal Rationale.** We for the first time introduce the rationale element in ABSA, enhancing the definition by providing deeper insights into the motivations behind sentiments, allowing an interpretable sentiment understanding at a cognitive level.

**Dynamic Sentiment Flipping.** Going beyond the traditional ABSA benchmark, PanoSent pioneers the examination of sentiment flips, studying the dynamics nature of ABSA.

**Multi-scenario.** PanoSent takes the dialogue as the context backbone, covering 10 main real-life domains across over 100 subdomains, ensuring an extensive diversity that supports research into sentiment analysis from various perspectives.

**Multimodality.** Beyond textual content (56%), PanoSent comprises three other modalities of information, including images (22%), audio (6%), video (4%), and mixed modalities (12%).

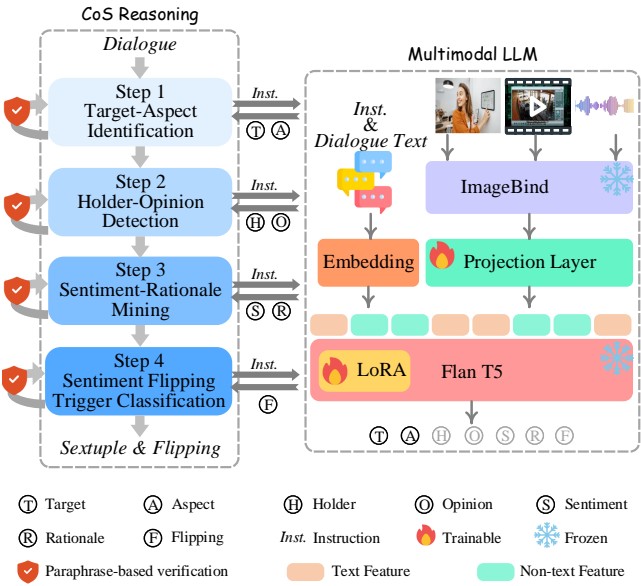

**Figure 2: Schematic overview of our Sentica MLLM.**

**Multilingualism.** PanoSent covers three mainstream languages, English (60%), Chinese (30%), and Spanish (10%), allowing a cross-lingual study of ABSA.

**Implicit ABSA.** Our dataset fully supports implicit ABSA, thereby elevating the challenges. While most of the sextuples are explicitly mentioned in the utterance text, 28% of the sextuples contain elements that need to be implicitly inferred from contexts or various modality information.

**High-quality and Large-scale.** Through careful manual annotation and cross-validation, we ensure the high quality of PanoSent. By employing automated synthesis, we significantly expand the scale of the dataset without compromising its quality, resulting in a total of 10,000 dialogue instances and 47,100 sextuples. The statistics are presented in Table 2.

## 5  Methodology

The two tasks in PanoSent encompass non-trivial challenges, e.g., complex conversational context understanding, multimodal feature extracting, and cognitive-level ABSA reasoning. To address these, we propose a comprehensive solution. Below, we detail the models proposed, the reasoning framework, the verification mechanism, and the learning approach.

### 5.1  Multimodal LLM Backbone

Currently, LLMs demonstrate remarkable capabilities in understanding language semantics. Correspondingly, MLLMs have been developed, exhibiting powerful abilities to comprehend multimodal data [39]. Building on the success of MLLMs, we consider leveraging them to help solve our task, where a thorough understanding of multimodal information is required. To this end, we develop a novel MLLM, **Sentica**, as presented in Figure 2. We adopt the Flan-T5 (XXL) [11] as the core LLM for semantics understanding and decision-making. Besides texts that are directly input into LLM, for non-text inputs, we adopt multimodal models to encode the

signals into the LLM-understandable representations. Specifically, we leverage ImageBind [26] as the unified encoders for all three non-text modalities, due to its prominent ability. Then, a linear layer connects ImageBind to LLM for representation projection.

### 5.2  CoS Reasoning Framework

Resolving two tasks, Panoptic Sentiment Sextuple Extraction and Sentiment Flipping Analysis, is challenging, not only due to the complex task definitions but also the cognitive-level requirement on the causal rationale and trigger detection. Inspired by the recent Chain-of-Thought (CoT) reasoning paradigm [73], here we also consider a human-like process of sentiment understanding and propose a Chain-of-Sentiment (CoS) reasoning framework. Previous ABSA studies [18] reveal that various ABSA elements can play hierarchical roles in depicting the overall sentiment puzzle. For example, the opinion should be detected before determining the sentiment polarity; likewise, identifying the target and aspect has a higher priority over recognizing the opinion. Thus, our main idea is that we deconstruct the two subtasks into four progressive, chained reasoning steps, from simpler to more complex. Using the capability of Sentica, solving each step incrementally accumulates key clues and insights for the follow-up steps. Figure 2 also illustrates how the CoS reasoning works with Sentica.

**Step 1: Target-Aspect Identification.** Given input dialogue $D$ possibly with multimodal signals and with specific instruction $P_1$, the initial step aims to prompt Sentica to identify all the possible **targets** and their specific **aspects** discussed within the dialogue, i.e., $\{(t_i, a_i)\}$.

---

**Input Data**: $D$
**Instruction**: Based on the multi-party dialogue and its accompanying multimodal data, please identify all possible targets and their specific aspects mentioned in the dialogue. Extract each target and aspect explicitly from the utterance text spans, or infer them implicitly via your understanding of the input data. Ensure each identified target is paired with its aspect(s), forming target-aspect pairs.

**Expected Output**: (target, aspect)$_1$, (target, aspect)$_2$, $\cdots$

---

This step can be formulated as:

$$\{(t_i, a_i)\} \leftarrow f_1(D|P_1). \tag{1}$$

**Step 2: Holder-Opinion Detection.** The second step is to detect the **holders** $h_j$ and their specific **opinions** $o_j$, regarding the identified targets and aspects. We require Sentica to output a set of quadruples consisting of the holder, target, aspect, and opinion $\{(h_j, t_i, a_i, o_j)\}$. After this step, we construct holder-target-aspect-opinion quadruples, which lay the foundation for understanding the further sentiment.

---

**Input Data**: $D, \{(t_i, a_i)\}$
**Instruction**: Based on the dialogue and each target-aspect pair identified previously, please identify the holder (the person who expresses an opinion, normally should be a speaker of certain dialogue utterance) and the opinion, both either directly extracted from the text or inferred from our understanding of the input data. Formulate your output into 'holder-target-aspect-opinion' quadruples, ensuring each element is clearly identified.

**Expected Output**: (holder, target, aspect, opinion)$_1$, (holder, target, aspect, opinion)$_2$, $\cdots$

---

**Table 3: Main results of Subtask-I, Panoptic Sentiment Sextuple Extraction. 'H/T/A/O/R/S' represents Holder, Target, Aspect, Opinion, Rationale, and Sentiment, respectively. All the scores are averaged over five runs under different random seeds.**

| | | Model | PLM | Element-wise | | | | | Pair-wise | | | | Sextuple | |
|---|---|---|---|---|---|---|---|---|---|---|---|---|---|---|
| | | | | H | T | A | O | R | T-A | H-O | S-R | O-S | Micro | Iden. |
| **EN** | M1 | DiaASQ | mBERT Base | 69.56 | 58.61 | 52.04 | 44.39 | 22.90 | 33.07 | 33.52 | 18.98 | 40.26 | 13.49 | 19.07 |
| | M2 | UGF | mT5-XXL | 71.17 | 61.83 | 55.25 | 47.68 | 25.87 | 35.39 | 36.08 | 22.37 | 42.80 | 15.85 | 20.12 |
| | M3 | Unified-IO 2 | Unified-IO 2 7B | 75.82 | 65.81 | 59.50 | 51.57 | 29.03 | 39.41 | 40.36 | 26.16 | 47.03 | 18.95 | 22.03 |
| | M4 | NExT-GPT | Vicuna 7B | 76.07 | 66.25 | 59.97 | 52.12 | 29.95 | 40.23 | 41.24 | 27.07 | 47.89 | 20.01 | 24.98 |
| | M5 | Sentica | Flan-T5-XXL | 77.48 | 67.49 | 61.01 | 53.06 | 31.02 | 41.12 | 42.31 | 28.12 | 48.94 | 21.26 | 25.67 |
| | M6 | Sentica (+CoT) | Flan-T5-XXL | 80.98 | 72.85 | 67.21 | 58.07 | 38.10 | 46.49 | 47.35 | 34.47 | 55.25 | 26.69 | 30.95 |
| | M7 | Sentica (+CoS) | Flan-T5-XXL | 83.41 | 75.70 | 70.38 | 60.96 | 41.35 | 49.72 | 50.47 | 37.27 | 58.20 | 29.71 | 33.69 |
| | M8 | Sentica (+CoS+PpV) | Flan-T5-XXL | **84.30** | **76.51** | **71.16** | **62.47** | **43.23** | **51.09** | **52.20** | **39.50** | **60.25** | **32.18** | **35.72** |
| **ZH** | M9 | DiaASQ | mBERT Base | 66.02 | 55.07 | 50.66 | 40.21 | 18.19 | 29.33 | 30.90 | 16.15 | 37.89 | 11.05 | 16.25 |
| | M10 | UGF | mT5-XXL | 67.81 | 57.86 | 53.72 | 43.15 | 21.17 | 31.71 | 33.49 | 18.63 | 39.88 | 13.70 | 17.09 |
| | M11 | Sentica | ChatGLM2 6B | 74.19 | 64.20 | 58.45 | 49.39 | 28.04 | 38.02 | 38.16 | 24.61 | 45.70 | 18.57 | 22.86 |
| | M12 | Sentica (+CoT) | ChatGLM2 6B | 77.76 | 68.82 | 64.21 | 54.43 | 34.70 | 42.87 | 43.23 | 30.69 | 51.58 | 23.64 | 27.88 |
| | M13 | Sentica (+CoS+PpV) | ChatGLM2 6B | **80.05** | **72.29** | **67.83** | **58.25** | **38.96** | **46.82** | **48.04** | **35.78** | **56.61** | **28.06** | **31.91** |
| **SP** | M14 | DiaASQ | mBERT Base | 63.72 | 53.80 | 46.33 | 36.59 | 17.02 | 26.89 | 29.61 | 14.52 | 35.13 | 8.23 | 13.68 |
| | M15 | UGF | mT5-XXL | 65.14 | 55.69 | 49.17 | 39.57 | 19.89 | 29.44 | 31.02 | 16.03 | 37.06 | 11.11 | 14.92 |
| | M16 | Sentica | Vicuna 7B | 71.61 | 62.02 | 55.83 | 47.02 | 25.73 | 35.77 | 35.83 | 22.17 | 43.04 | 15.97 | 20.12 |
| | M17 | Sentica (+CoT) | Vicuna 7B | 74.89 | 66.34 | 61.83 | 51.94 | 32.51 | 40.26 | 40.88 | 28.07 | 48.84 | 21.16 | 25.40 |
| | M18 | Sentica (+CoS+PpV) | Vicuna 7B | **77.49** | **69.85** | **65.31** | **55.62** | **36.66** | **44.37** | **45.54** | **33.39** | **54.05** | **25.62** | **29.54** |

This step is formulated as:

$$\{(h_j, t_i, a_i, o_j)\} \leftarrow f_2(D, \{(t_i, a_i)\}|P_2). \quad (2)$$

**Step 3: Sentiment-Rationale Mining.** The third step then analyzes the **sentiment** $s_k$ with each opinion and identifies the **rationale** $r_l$, based on the identified holder-target-aspect-opinion quadruples. We ask Sentica to output a set of sextuplets, by further adding sentiment and rationale to the previous quadruples to form $\{(h_j, t_i, a_i, o_j, s_k, r_l)\}$.

> **Input Data**: $D, \{(h_j, t_i, a_i, o_j)\}$
> **Instruction**: Based on the dialogue and each holder-target-aspect-opinion quadruple identified previously, please identify the sentiment polarity associated with the opinion and analyze the causal rationale behind it. The sentiment polarity should be classified as 'positive', 'neutral', or 'negative'. The rationale should be extracted explicitly from the text, or inferred implicitly via your understanding of the input data. Formulate your output into 'holder-target-aspect-opinion-sentiment-rationale' sextuples, ensuring sentiment polarity is clearly analyzed and the other five elements are clearly identified.
>
> **Expected Output**: (holder, target, aspect, opinion, sentiment, rationale)$_1, \cdots$

We denote this step as:

$$\{(h_j, t_i, a_i, o_j, s_k, r_l)\} \leftarrow f_3(D, \{(h_j, t_i, a_i, o_j)\}|P_3). \quad (3)$$

**Step 4: Sentiment Flipping Trigger Classification.** With all the sextuplets detected, the final step of discerning sentiment flipping would be much effortless. Specifically, we prompt Sentica to first summarize any changes (i.e., from an **initial sentiment** ($\zeta_k$) to a **flipped sentiment** ($\phi_k$)) in sentiment of same *holder-target-aspect*, and then classify the **trigger** ($\tau_m$) label for each sentiment flip.

The output is a set of sextuplets: $\{(h_j, t_i, a_i, \zeta_k, \phi_k, \tau_m)\}$.

> **Input Data**: $D, \{(h_j, t_i, a_i, o_j, s_k, r_l)\}$
> **Instruction**: Based on the dialogue and each holder-target-aspect-opinion-sentiment-rationale sextuple, please identify instances where a sentiment flip occurs for the same holder regarding the specific target-aspect pair. Determine the trigger type for these flips from the predefined categories: *introduction of new information*, *logical argumentation*, *participant feedback and interaction*,

> *personal experience and self-reflection*. Formulate your output to include the holder, target, aspect, initial sentiment, flipped sentiment, and the trigger type, or state "None" if no flips are identified.
>
> **Expected Output**: (holder, target, aspect, initial sentiment, flipped sentiment, trigger type)$_1, \cdots$; or "None"

This step can be marked as:

$$\left\{ \begin{matrix} \text{NONE}, & \text{if no flip} \\ (h, t, a, \zeta, \phi, \tau), & \text{if flip} \end{matrix} \right\} \leftarrow f_4\left(D, \{(h_j, t_i, a_i, o_j, s_k, r_l)\}|P_4\right). \quad (4)$$

### 5.3 Paraphrase-based Verification

Given that we designed the entire two-task solution as a step-wise process, a potential issue is that CoS could lead to error accumulation. For example, an error in the first step could directly impact the outcome of all subsequent steps. Therefore, it's crucial to perform verification at every reasoning step. Existing work has verified that compared to structured data, LLMs excel more in understanding natural language [36, 69]. This implies that having LLMs directly check the correctness of each obtained $k$-tuple is sub-optimal. A more intuitive approach is to first convert the structured $k$-tuples into natural language expressions through paraphrasing, effectively creating a claim that conveys the same meaning in a different format. Then, let the LLM check whether this claim is in an entailment or contradiction relationship [34, 65] with the given dialogue context and information. We refer to this as a *Paraphrase-based Verification* (PpV) mechanism. If the relationship is one of entailment, the verification is successful, and the process moves on to the next reasoning step. If it's a contradiction, the current step is rerun until a reasonable result is yielded. This process not only ensures that each reasoning step is built on verified information but also enhances the overall robustness of sentiment analysis, effectively mitigating the negative impact of hallucinations [31, 64] inherent in LLMs.

### 5.4 Instruction Tuning

To empower Sentica with the reasoning capabilities of the CoS framework, we conduct instruction tuning, entailing a three-phase training process. In the first stage, we enable the LLM to understand multimodal representations bound to images, audios and videos. We consider training directly on existing 'text-X' pair datasets (where

**Table 4: Results of the Subtask-II, Sentiment Flipping Analysis.**

| | Model | EN | | | ZH | | | SP | | |
|---|---|---|---|---|---|---|---|---|---|---|
| | | Flip | Trig | Flip-Trig | Flip | Trig | Flip-Trig | Flip | Trig | Flip-Trig |
| M1 | NExT-GPT | 60.27 | 63.43 | 55.80 | / | / | / | 51.32 | 55.52 | 46.02 |
| M2 | Sentica | 63.71 | 66.26 | 58.49 | 58.83 | 62.50 | 52.57 | 55.37 | 59.61 | 50.98 |
| M3 | Sentica (+CoT) | 65.53 | 69.30 | 61.99 | 61.79 | 65.70 | 58.04 | 58.31 | 62.57 | 55.28 |
| M4 | Sentica (+CoS) | 69.89 | 73.25 | 66.06 | 65.91 | 69.67 | 62.35 | 62.24 | 66.66 | 59.40 |
| M5 | Sentica (+CoS+PpV) | **72.57** | **76.18** | **69.39** | **68.68** | **72.41** | **65.46** | **65.75** | **69.45** | **62.52** |

'X' refers to image, audio, or video), i.e., inputting 'X' and having the LLM output the corresponding caption text. In the second stage, we aim for the LLM to smoothly and accurately execute the CoS reasoning process. We consider using the PanoSent train set as supervised data, wrapping the corresponding CoS instructions to obtain instruction fine-tuning data. Then, we train the model on the data to master the response mode for the corresponding inputs and outputs. The third stage teaches Sentica the PpV pattern. Based on the previous CoS instructions, we construct correct verification pairs with an entailment relation. Meanwhile, by arbitrarily altering elements of the $k$-tuple, we create contradictory relations in paraphrases as counterexamples, on which we fine-tune Sentica.

## 6 Experiments

### 6.1 Settings

**Evaluations.** For Task-I, we follow DiaASQ [37], considering evaluation under three dimensions: 1) element-wise detection; 2) pairwise extraction; 3) overall sextuple extraction. For the explicit elements, we use the *exact match* F1 metric. For the implicit elements, we use the *binary match* F1, where we use GPT-4 to evaluate if the gold element is semantically identical to the prediction (1 if yes, otherwise 0). Since a correct rationale element may not need to strictly match gold term boundaries (i.e., only coinciding with the critical part), we take the *proportional match* F1 for its evaluation. For the compound evaluation, a pair or overall sextuple is correct only when all elements are correct. Here, the score for rationale above 0.5 is deemed a correct prediction. For sextuple extraction, *micro F1* evaluates the entire sextuple, while *identification F1* measures the sextuple without sentiment polarity. For subtask-II, we mainly measure three targets: 1) if both Initial Sentiment & Flipped Sentiment (Flip) are correct, 2) if the flipping trigger's category (Trig) is correct, and 3) if both Flip-Trig is correct simultaneously. For (1) and (3), we use *exact match F1*; for (2), we adopt *macro F1*.

**Baselines.** Since no prior research or methods can be directly adopted here for comparisons, we consider maintaining several baselines via our implementations. We first retrofit the UGF [81] and DiaASQ [37] so that they can execute the multimodal sextuple extraction tasks, where the small-size LMs are used, e.g., Multi-lingual BERT (Base) [14] and mT5 (XXL) [80]. We also consider existing MLLMs (supporting T/A/I/V) for comparisons, including Unified-IO 2 [48] and NExT-GPT [76]. All systems are fine-tuned using the PanoSent training set for fairness.

**Implementations.** Given the varying capabilities of different LLMs across languages, we use Flan-T5 (XXL) for English data, ChatGLM2 6B for Chinese data, and Vicuna 7B for Spanish data. Our Sentica is tuned via LoRA [28], allowing for the least parameter updating. The experiments were conducted on hardware with 8*A100 GPUs,

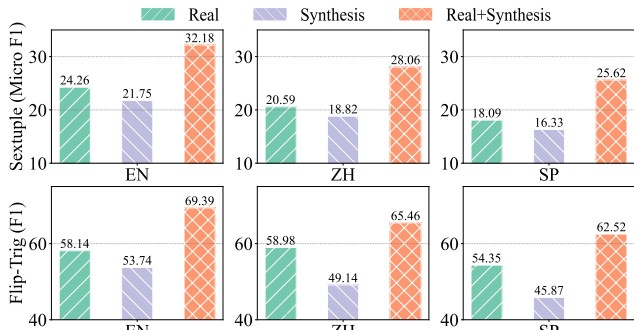

**Figure 3: Performance with different data sources.**

each boasting 80GB of memory. To ensure the reliability and reproducibility of our results, we tune the system on a developing set and used five different random seeds, selecting our experimental outcomes based on the average scores from five runs.

### 6.2 Main Results

**Performances on Panoptic Sentiment Sextuple Extraction.** Table 3 compares the performances of different methods on Subtask-I, where we can gain the following observations. First, due to the presence of many implicit elements in our data, the performance of extraction-based baselines (such as DiaASQ and UGF) can be inferior. The generative nature of LLM-based methods, however, effectively addresses this, resulting in overall better performance. Comparing the performance of Sentica with Unified-IO 2 and NExT-GPT (M3&4 vs. M5), we see that our method performs better. Sentica, when equipped with the CoS framework, shows significant improvement over the direct prompting paradigm (M7 vs. M5). Moreover, comparing M6 and M7 shows a clear advantage of our proposed CoS reasoning framework over the vanilla CoT method. Most importantly, when Sentica combines both the CoS and PpV mechanisms, the complete system (M8) exhibits the strongest global performance. As seen, across different task evaluation granularities and languages, our system achieves the best scores. In both ZH and SP languages, our system also demonstrates a significant superiority over the Sentica CoT-based variant. Finally, we can observe task evaluation from different perspectives. For different elements, the identification of the holder and target is more accurate, while the determination of rationale is more challenging. Similarly, the recognition of sentiment-rationale pairs is also more difficult. The overall identification of sextuples poses the greatest challenge, providing a challenging benchmark for follow-up research.

**Results on Sentiment Flipping Analysis.** For Task 2, we present the overall results in Table 4. Similar trends to those observed in

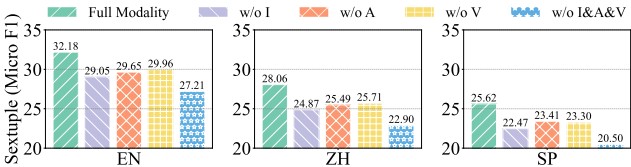

Figure 4: Evaluation of the contribution of each modality.

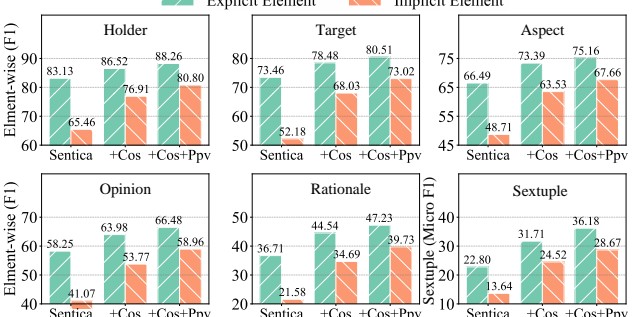

Figure 5: Analysis of Explicit and Implicit Elements.

Table 3 are evident. For instance, our Sentica, on the same backbone LLM, outperforms NExT-GPT. Additionally, the CoS reasoning approach, compared to direct prompting or the CoT technique, significantly enhances the accuracy of sentiment flipping identification across all languages. Moreover, our complete system (i.e., Sentica+CoS+PpV) demonstrates the best performance. The main results and observations from the above two subtasks evidently demonstrate the effectiveness of our proposed methods.

## 6.3 Analysis and Discussion

We take one step over the overall performance, further delving into the analyses of the proposed data and methods.

**Q1: Is It Necessary to Construct Synthetic Data?** In the above experiments, we train the model by combining real data with synthetic data. Therefore, we plan to train the model using these two types of data separately and compare the performance. The results for the two subtasks under different languages are shown in Figure 3. Overall, it is observable that training on real-life data yields better results compared to training on synthetic datasets, even though the latter are more plentiful. This is because real data possess a more authentic distribution of information, enabling the model to learn a richer set of features. Moreover, our test set is also sampled from real data. Most importantly, we discover that once synthetic data is used as an additional supplement to substantially expand the quantity of real data, it can significantly enhance the final performance, consistently. This proves the necessity to construct synthetic data.

**Q2: How Significant Is the Role of Multimodal Information?** Although multimodal information has been utilized in existing multimodal sentiment analysis research [35, 53], it is mostly treated as supplementary to textual information for aiding in the determination of sentiment polarity. In this work, the role of multimodal information is comprehensive and all-encompassing. It not only assists in determining sentiment polarity but also serves as a direct source of information for judging the sextuple elements (i.e., in an implicit manner). We demonstrate the impact of removing

Table 5: Comparison of different verification mechanisms.

| | Task F1 | | Human Acc. |
|---|---|---|---|
| | Sextuple | Flip-Trig | Entail Detect. |
| PpV (paraphrase via template) | **32.18** | **69.39** | **88.15** |
| PpV (paraphrase via LLM) | 30.83 | 67.60 | 73.62 |
| dir. verify | 30.26 | 67.04 | / |
| w/o verify | 29.71 | 66.06 | / |

multimodal information from the test set on the performance of the sextuple extraction task in Figure 4. As seen, removing any type of modal signal results in a downgrade in performance, with the information from images being the most crucial. Removing all non-text modalities has the most significant impact.

**Q3: How Are Performance for Explicit and Implicit Elements Individually?** We define the sextuple extraction wherein elements can either be explicitly derived from text or implicitly inferred from contexts or various modalities. While the overall results previously presented combine the performance of both explicit and implicit elements, here we aim to showcase the specific performance of various elements individually. As presented in Figure 5, the performance of implicit elements is consistently lower than that of explicit elements. This indicates that recognizing implicit elements is much more challenging. This phenomenon aligns with reality; because, compared to extracting explicit text, identifying implicit elements requires a comprehensive understanding of the context's semantics before inferring the corresponding elements.

**Q4: Is the PpV Mechanism Reasonable?** Lastly, we verify the rationality of the proposed PpV mechanism. We adopt a template-based approach for paraphrasing *k*-tuples, then check whether the semantics of the structured data coincide with the given context of dialogue. In Table 5, we present some evaluations. We explore the task performance under different mechanisms, including paraphrasing via LLM, direct verification without paraphrasing, and without any verification. It is evident that the PpV mechanism outperforms both direct verification and no verification. Furthermore, for PpV, we conduct entailment detection between the obtained paraphrases and the dialogue context through human evaluation and then report the accuracy. We see that using fixed templates for paraphrasing is more reliable than utilizing LLMs to paraphrase structured tuples.

## 7 Conclusion

This paper introduces a novel multimodal conversational ABSA, where the Panoptic Sentiment Sextuple Extraction (including holder, target, aspect, opinion, sentiment, and rationale) and the Sentiment Flipping Analysis tasks are proposed, providing a comprehensive and panoptic definition of sentiment analysis that aligns with the complexity of human-level emotional expression and cognition. We benchmark the novel settings with PanoSent, a large-scale high-quality dataset annotated both manually and automatically, featuring conversational contexts, multimodality, multilingualism, and multi-scenarios. We then benchmark the tasks with an effective Chain-of-Sentiment reasoning framework, together with a novel MLLM (namely Sentica) and a paraphrase-based verification mechanism, serving as a strong baseline for subsequent research.

# Acknowledgments

This work is supported by the Ministry of Education, Singapore, under its MOE AcRF TIER 3 Grant (MOE-MOET32022-0001).

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
