# OpenReview forum: "PanoSent: A Panoptic Sextuple Extraction Benchmark for Multimodal Conversational Aspect-based Sentiment Analysis"
_acmmm.org/ACMMM/2024/Conference — MM2024 Oral_

### Official Review · Reviewer_xaWk · 2024-05-14

**Rating:** 5
**Confidence:** 3

**Summary:**

The paper introduces a novel benchmark, PanoSent, for multimodal conversational aspect-based sentiment analysis. It addresses gaps in existing research by proposing two subtasks: Panoptic Sentiment Sextuple Extraction and Sentiment Flipping Analysis. The authors propose a dataset annotated through a combination of human annotation and auto-synthesis methods, ensuring high quality and comprehensive coverage. Key features include a fine-grained sentiment definition, cognitive causal rationale exploration, dynamic sentiment flipping analysis, coverage of multiple real-life scenarios, incorporation of various modalities such as text, images, audio, and video, support for multiple languages, and inclusion of implicit aspect-based sentiment analysis elements. The dataset, with its large-scale and high-quality data, serves as a valuable resource for researchers aiming to delve deeper into sentiment analysis and cognitive reasoning in complex conversational settings. The PanoSent benchmark aims to advance sentiment analysis by providing a panoramic view of sentiments in diverse conversational contexts.

**Strengths:**

Following are the main strengths of the paper:

Comprehensive Definition: Introducing a more comprehensive definition of aspect-based sentiment analysis at a cognitive level, the paper defines Panoptic Sentiment Sextuple Extraction and Sentiment Flipping Analysis tasks, offering a detailed view of sentiments in conversational contexts.

High-Quality Benchmark Dataset: The construction of the PanoSent dataset, with annotations from human experts and auto-synthesis methods, ensures high quality and extensive coverage. The dataset spans multiple domains, modalities, languages, and scenarios, providing a rich resource for sentiment analysis research.

Innovative Reasoning Framework: The paper proposes the Chain-of-Sentiment reasoning framework, coupled with the Sentica Multimodal Large Language Model, to achieve high task performance. This framework breaks down the sentiment analysis task into progressive reasoning steps, enhancing the extraction of sentiment elements and identification of flipped sentiments with corresponding rationales and triggers.

**Limitations:**

Dataset Building and Annotation: While auto-synthesis is employed to augment the dataset and reduce manual annotation efforts, there may be instances where synthesized data poses challenges in terms of quality and reliability. How do the authors deal with this? Were there specific annotation guidelines given for human annotators during recruitment, considering that ABSA-based annotation is quite challenging in itself? Additionally, samples from the annotated dataset are missing.

Qualitative Analysis: The dataset contains sextuples with implicit elements that require inference from context or modalities. Recognizing and extracting implicit sentiments poses a significant challenge, as it necessitates a deep understanding of the semantic context before inferring corresponding sentiment elements. Test samples comparing the performance of the proposed model with other best-performing baselines could be insightful.

Error Analysis/Limitations: What are the challenges faced and possible future directions regarding the methodology?

**Suitability:**

3

---

### Official Review · Reviewer_KXxx · 2024-05-15

**Rating:** 4
**Confidence:** 3

**Summary:**

This paper proposes a new benchmark called PanoSent for multimodal conversational aspect-based sentiment analysis (ABSA). It introduces two novel subtasks: 1) Panoptic Sentiment Sextuple Extraction, which recognizes holders, targets, aspects, opinions, sentiments, and rationales from multi-turn multimodal dialogues, and 2) Sentiment Flipping Analysis, which detects dynamic sentiment changes and their causal reasons. The PanoSent dataset contains 20,000 annotated dialogues, covering multiple modalities, languages (English, Chinese, Spanish), and domains. A Chain-of-Sentiment reasoning framework and Sentica multimodal large language model are developed to address the tasks effectively.

**Strengths:**

1.Novelty and Relevance: The paper addresses a significant gap in multimodal conversational ABSA by introducing innovative tasks that extend beyond traditional sentiment analysis. The integration of multimodal data and conversational contexts in sentiment analysis is both timely and relevant, given the increasing prevalence of multimedia communication.
2.Comprehensive Dataset: The construction of the PanoSent dataset is a major strength. It is one of the first to include such a wide range of modalities (text, image, audio, video), languages (English, Chinese, Spanish), and scenarios (over 100 domains). This diversity supports the development and testing of ABSA systems in realistic, varied conditions.
3.Technical soundness: The proposed Chain-of-Sentiment reasoning framework and Sentica MLLM are well-motivated and technically sound. Breaking down the complex tasks into progressive reasoning steps and leveraging the power of MLLMs for multimodal understanding are effective approaches. The paraphrase-based verification further enhances robustness.
4.Clarity and Presentation: The paper is well-organized and clearly written. The introduction of complex concepts is handled adeptly, making the research accessible to readers with varying levels of expertise in sentiment analysis.

**Limitations:**

1.In lines 152-156, you mention extending the ABSA quadruple extraction to include the holder and rationale, claiming this covers all fine-grained emotional elements. However, the criteria for choosing these components are not fully clear. Other elements like sentiment intensity could also be considered. Providing citations to support how these 6 elements comprehensively capture sentiment would be helpful. The holder and rationale enable Task 2, but may not necessarily represent an exhaustive set of sentiment elements.
2.Lines 311-314 introduce 4 pre-defined categorical labels for sentiment change reasons. Could you please clarify the basis for selecting these specific categories? Are they grounded in existing research or empirical observations? Additional context would be useful here.
3.The term "Vanilla CoT" in line 743 could benefit from further explanation and context to avoid appearing vague. Providing more details on how CoS improves upon CoT would strengthen this discussion.
4.In section 5.2, the CoS method seems to apply the CoT approach to ABSA. To support the claim that CoS outperforms CoT, it would be valuable to include an example of CoT, perhaps in an appendix. Without this, it's harder to assess whether the CoS design choices, rather than suboptimal CoT prompts, drive the performance differences.
5.There are a few definitive statements that would be bolstered by additional citations or justification, as mentioned above.

**Suitability:**

3

---

### Official Review · Reviewer_Z6fE · 2024-05-24

**Rating:** 5
**Confidence:** 3

**Summary:**

This paper introduces Panosent- a new multimodal, multilingual benchmark dataset to extend the research focussed in the area of Aspect based sentiment analysis and proposes two new subtasks- such as Panoptic Sentiment Sextuple Extraction and Sentiment Flipping Transition analysis. In addition to this, they construct a new framework termed as Chain-of-Sentiment reasoning integrated deep into the
Aspect-based Sentiment Analysis and construct a multimodal architecture termed Sentica leveraging this framework.

**Strengths:**

- The proposed benchmark dataset Panosent- differs from the previously proposed datasets in the field of Aspect based sentiment analysis in that it introduces the aspect of Casual rationality, Dialogue Sentiment transition labels through the contribution of Holder, Rationale target aspects (both explicit and 28% implicit elements) and adds another multimodal video dimension spanning English, Chinese and Spanish languages collected from Social Media (Twitter, 325 Facebook, Reddit, Weibo, Xiaohongshu, BeReal, etc..)
- Chain of Sentiment Framework- Leveraging the hierarchical roles that the intertwined targets within the aspect based sentiment analysis play, the authors develop  four progressive, reasoning steps chained together with increasing complexity
- To avoid catastrophic error propagation within the unsupervised chain of Sentiment framework, the authors introduce paraphrase based verification utilizing the fact that LLMs excel reasoning over composed Natural language which is utilized in the third/final stage of the training process, followed by the text-multimedia pair and CoS training.

**Limitations:**

- The work puts together a self verified chain of thought reasoning framework tailored with adaptations to that of the Aspect based Sentiment systems as Chain of Sentiment framework.  Further the chain of sentiment framework is combined with a multimodal fusion network methodology which binds together and attends to multimodal representations of the proposed dataset samples.
- In the multilingual evaluation section, especially for the Chinese and Spanish cases, the ablation of Sentica+CoS with standalone CoS is missing, it would be interesting to have those results as it would help us to note how much the standalone Sentica+CoS topology can outperform other baselines in a multilingual setting without the addition of the paraphrase verification step which relies on accurate entailment or contradiction response of the k-tuple targets.
- It would be interesting to have evaluations of other dialogue based instruction tuned models along with Chain of Thought reasoning as baselines and make more relevant comparisons to the proposed dataset, be it instruction tuned LLMs or VLMs as applicable .
- Though the CoS framework's prompt instructions are clearly spelled, it would be interesting if the authors could note if there happen to be any more fundamental differences in how different the CoS functions as opposed to the CoT framework apart from the inherent hierarchy built out of the ASBA area of research.
- Also, the authors note that the training involved to be a three stage process and denote the trainable parameters in the LoRA and projection layers, it would also be interesting if the authors could further enumerate the association of trainable parameters with each stage of the training process.
- Synthetic generation part of the dataset remains unclear, if he work could offer additional details on pseudo dialogues generation mechanism and the strategy of how the synthetic part could be seamlessly integrated and concatenated to the real dataset in hand that would help too.

**Suitability:**

3

---

### Meta-Review · Area_Chair_LBEp · 2024-06-30

**Recommendation:** Accept (Oral)
**Confidence:** 5

**Metareview:**

The paper introduces PanoSent, a novel benchmark for multimodal conversational aspect-based sentiment analysis, addressing significant gaps in existing research with two innovative subtasks: Panoptic Sentiment Sextuple Extraction and Sentiment Flipping Analysis. The dataset, annotated through human and auto-synthesis methods, spans multiple modalities, languages, and domains, offering a comprehensive resource for sentiment analysis research. The Chain-of-Sentiment reasoning framework and Sentica multimodal large language model demonstrate technical soundness and efficacy in task performance. However, concerns remain regarding the quality and reliability of synthesized data, the explicit criteria for extending ABSA, and the basis for predefined sentiment change categories. Comparative performance analysis with the latest multimodal models and detailed error analysis are needed. Additionally, clarifying the synthetic generation process and enhancing the multilingual evaluation would strengthen the work. Addressing these reviewer concerns in the camera-ready version will enhance the robustness and applicability of the proposed benchmark and methodology.